# Evaluation of Surface Topography after Face Turning of CoCr Alloys Fabricated by Casting and Selective Laser Melting

**DOI:** 10.3390/ma13112448

**Published:** 2020-05-27

**Authors:** Marta Beata Krawczyk, Marcin Andrzej Królikowski, Daniel Grochała, Bartosz Powałka, Paweł Figiel, Szymon Wojciechowski

**Affiliations:** 1Department of Manufacturing Engineering, Faculty of Mechanical Engineering and Mechatronics, West Pomeranian University of Technology Szczecin, Ave. 19, Piastow, 70-310 Szczecin, Poland; makro@zut.edu.pl (M.A.K.); daniel.grochala@zut.edu.pl (D.G.); bartosz.powalka@zut.edu.pl (B.P.); 2Department of Materials Technology, Faculty of Mechanical Engineering and Mechatronics, West Pomeranian University of Technology Szczecin, Ave. 19, Piastow, 70-310 Szczecin, Poland; pfigiel@zut.edu.pl; 3Institute of Mechanical Technology, Faculty of Mechanical Engineering and Management, Poznan University of Technology, Piotrowo 3, 60-965 Poznan, Poland; szymon.wojciechowski@put.poznan.pl

**Keywords:** surface topography, chrome-cobalt alloy, SLM, face

## Abstract

The machinability of hard-to-cut CoCr alloys manufactured by Selective Laser Melting (SLM) technology is not yet sufficiently studied. Therefore, this work focuses on evaluation of surface texture formation during face turning of CoCr alloy. As part of the research, two specimen types were subject to comparison: made with the application of conventional casting and manufactured by additive manufacturing—SLM. A number of analytical and experimental methods were employed to describe the specimen composition and morphology, as: X-Ray Diffraction Analysis (XRD), optical metallurgical microscopy, confocal optical microscopy, and Vickers hardness HV_0.1_ measurements. In the next stage, the measurements of surface topographies formed during turning in a range of variable cutting speeds and feeds were carried out. Ultimately the multi-factor MANOVA (Multivariate Analysis of Variance) illustrating the influence of manufacturing technology, cutting speed, and feed ratio on selected surface parameters of samples was made. It has been demonstrated that during face turning with feeds up to 0.15 mm/rev, the similar values of surface roughness height and material ratio curve parameters were reached for both tested CoCr alloys. However, in a range of higher feed values, the surface quality of CoCr samples fabricated by SLM was lower than that reached for CoCr after casting process.

## 1. Introduction

Cobalt chromium alloys are acknowledged as attractive materials in the application of many engineering fields. Shokrani et al. [1] listed applications of these materials in aero-engine and gas turbine parts due to their good corrosion resistance and high temperature strength. Limmahakhun et al. [2] noted that CoCr alloys apart from their high biocompatibility have greater strength and better wear resistance than Ti alloys, thus they are used in hip replacements. In general, the use of these materials for surgical implants took place over 60 years ago, which is emphasized by Gorardin et al. [3]. On the other side, Pieniążek et al. [4] indicated that the wide use of these alloys in biomedicine is due to the fact that they do not contain elements harmful for the human body, such as nickel or beryllium, which according to [5] allows for long-term incorporation in the human musculoskeletal system. However, next to orthopedic implantology, CoCr alloys are widely used in prosthetics and dental implantology. Due to main application area of CoCr alloys, i.e., medical implants, surface roughness is a key parameter. To obtain a good osteointegration with bones, rough surfaces are recommended. However, for surfaces exposed to biological fluids, a smooth, polished surface is required to minimize emerging bacterial habitats. For example, Takaichi et al. [6] and Dolgov et al. [7] studied the properties of CoCr alloys obtained by various techniques to apply them for removable partial dentures, metal frames, and porcelain-fused-to-metal crowns in dentistry, while Xin et al. [8] examined their release of toxic metal ions and the cell response. Ramirez-Ledesma et al. [5] noted that dental implant and/or prosthetic material must function mechanically without permanent deformation or failure while leaving the biological tissues completely unaltered. Dolgov [7] has revealed that good mechanical properties of the CoCr alloys, like high elasticity and hardness hinder their mechanical treatment. Zaman et al. [9] and Shao et al. [10] noted that low thermal conductivity of these alloys tends to increase the tool wear of the cutting tool due to rapid generation of high temperatures during the cutting process. In [1,9,10], the authors emphasize that machining of CoCr is commonly associated with short tool life due to the work hardening and attrition properties of the superalloy and poor surface finish due to severe surface damage of a machined workpiece resulting from heat generation and plastic deformations. Consequently, it can affect low productivity and high manufacturing costs. At this point it should be emphasized that when CoCr alloys are used in implantology, for example as a head of an artificial joint, the higher accuracy with surface finish is necessary to extend the life of the joint by compact abrasive wear and enhanced chemical stability. According to Jagtap [11] and Ketan [12], face turning, is, in particular, one of the important manufacturing processes producing the high accuracy and surface finish on metal implants. It is worth mentioning, according to Axinte [13] and Jagtap [14], that Co-Cr-Mo biocompatible alloys after face turning need a post finishing process, such as automated bonnet polishing, simultaneous five-axis polishing with resilient tools, drag finishing, magnetic abrasive finishing (MAF), or magneto-rheological fluid polishing for practical applications used in implant surfaces improvement (mirrored finish).

Cobalt chromium alloys are normally manufactured by the casting process, plastic working, forging process, and powder metallurgy technique [6,9]. Casted CoCr alloys have been widely used in the dentistry field and recently they are also used in the manufacturing of artificial joints [7]. The wrought alloy is utilized in the fabrication of stems of prostheses for heavy-loaded joints, such as the knee and hip [6,9]. In recent past years, an additive manufacturing technique that is selective laser melting (SLM) has been more and more widely used to produce biomedical parts, mainly in dentistry [3,8]. The SLM process could eliminate the disadvantages of the casting process, such as wax pattern distortion, irregularities in the cast metal, complicated procedures, and time-consuming processing [7]. The SLM enables the production of personalized complex objects and/or manufacturing of parts with dense structure and predetermined surface roughness. Moreover, it is a controllable, easy, and relatively quick process in comparison to classical fabrication processes [15]. According to Wang et al. [16], the mechanical properties of CoCr alloys fabricated using the SLM technology are strongly affected by the geometry of the molten pool. Therefore, the production of CoCr alloys with the high mechanical properties requires the effective selection of laser power, sweeping speed, and scanning line spacing. Due to improved mechanical performance, and better biocompatibility, compared to as-cast CoCr alloy a CoCr alloy fabricated by SLM is considered to be a potential candidate for bone implantation [2]. Allegri et al. [17] investigated machinability of medical grade Co-Cr-Mo alloy produced by selective laser melting during micromilling. Results showed that very low surface roughness can be achieved with the micro-milling operation; however, no correlation between surface roughness parameters (*Ra* and *Sa*) and feed per tooth was observed.

There is currently little research on the machinability of additively manufactured materials, in particular, the number of publications in the scope of surface topography formation for CoCr alloys during turning processes is very limited. In addition, these papers usually concern the cast materials [10,11,12,18]. On the other hand, the machinability of CoCr alloys produced with the application of the SLM technique is currently not recognized. Thus, this work proposes for the first time in the literature an evaluation of surface texture formation during face turning of cobalt-chromium alloys produced by the conventional casting and SLM method. The presented work focuses on the geometric part specification (GPS) parameters’ assessment of cobalt-chromium alloys obtained by the classic and additive manufacturing method (SLM).

In addition, the conducted research involves also the identification of the phases occurring in the tested cobalt-chromium alloys together with their crystallographic structure. The obtained results were used to confirm the phase identity of materials from two different manufacturing processes. Observations using optical microscopy were aimed at determining and comparing the microstructure of samples and testing their porosity. Microhardness tests were used to characterize and compare the mechanical properties of the tested materials. These materials were subjected to a face turning process, after which the GPS parameters were determined for the obtained surfaces. The obtained results provide for the first time important information regarding the surface texture formation during the cutting of CoCr alloys manufactured by SLM technology, as well as the recognition of these materials’ machinability, thereby enabling the selection of optimal cutting parameters.

## 2. Materials and Methods

Samples intended for the face turning tests made of the cobalt-chromium alloy were produced by two techniques: additive manufacturing and casting (the initial roughness parameters for the samples were respectively: Ra = 11.74 and 0.54 µm, and Rz = 70.73 and 4.16 µm). A series of samples with a diameter of 15 mm and a height of 30 mm were produced using the selective laser melting (SLM) on the REALIZER II 250 (MCP-HEK-REALIZER, Lübeck, Germany), equipped with a 100 W Nd: YAG laser. The SLM process parameters recommended by the powder manufacturer were used and are summarized in Table 1. The cylindrical samples with a diameter of 60 mm made by casting process have been supplied by Acnis International (France).

The phase composition of the tested materials was determined on the basis of X-Ray Diffraction Analysis (XRD). The PANalytical PW3040/60 X’Pert Pro apparatus equipped with a Cu Kα lamp (*λ* = 0.15406 nm) (Malvern Panalytical Ltd., Malvern, UK) operating at 40 kV and 35 mA was used in tests. Continuous scan mode of 2*Θ* was recorded in the range of 30° and 100° with a step size of 0.05° and counting time of 200 s per step. The microstructure tests were carried out on a NIKON EPIPHOT 200 metallographic microscope (Nikon, Tokyo, Japan). The microhardness tests in Vickers scale (HV0.1) were performed on the LECO LM247AT Microhardness Tester (Leco Corporation, St. Joseph, CT, USA) with Amh43 Software (version 1.51). For each of the samples, averaged measurements with at least 10 indentations, following ISO 6507-1:2018 Vickers hardness test [19], were made on the main diagonals with a load of 0.1 kg and a penetration time of 10 s. The distance between each indent was 2 mm.

The machining process was carried out on an AFM TAE 35 lathe (AFM DEFUM S.A., Andrychów, Poland). The CNMG 1204 08-SMR 1105 inserts made of tungsten carbide (WC) with PVD (Ti,Al)N_2_ coating intended for machining of difficult-to-cut materials were used. Recommendations for applications of cutting inserts manufacturer for hardly machinable (S) materials were followed. The basic parameters of the insert are listed in Table 2. The cutting process is shown in Figure 1.

Machining was carried out without cutting fluids at a variable cutting speed *v_c_* starting from 60 m/min, at a constant rotational speed *n* = 1250 min^−1^ for SLM samples and *n* = 312 min^−1^ for casting samples, the cutting depth *a_p_* = 0.5 mm. Dry machining enabled installation of the force gauge. Cutting forces were measured to monitor process stability. Information on cutting forces assured that the process was chatter free. The cutting speed was varied from the maximum value recommended by the cutting inserts manufacturer. Constant spindle speed was set in order to provide stable cutting conditions, i.e., chatter free machining. Cutting tests were preceded by impact tests of the tool and workpiece. Impact tests allowed us to evaluate dynamics at the tool tip. The dominant vibration mode, that may cause chatter instability has a frequency of *f_c_* = 104 Hz. The most stable speed was calculated according to formula *N = 60f_c_/k (k = 1*, *2*, *…).* Finally, we selected a spindle speed of 1250 RPM (close to 1248 RPM at *k = 5*) because it corresponds to the maximum recommended cutting speed for a given workpiece diameter. Tests were run at constant *a_p_* of 0.5 mm because such a value was significantly larger than the total height of the roughness profile *R_t_* of the samples fabricated by SLM. The cutting parameters are given in Table 3.

A new cutting edge was used for machining of each of the samples. Because of cutting speed variations in function of cutting path during the face turning, the radial cutting range was divided into 4 areas, for which an average constant speed was adapted in order to obtain data for statistical analyzes. Cutting speeds in selected areas were: V1 = max ***v_c_*** = 60 m/min, V2 = 0.75 max ***v_c_*** = 45 m/min, V3 = 0.5 max = 30 m/min, and V4 = 0.25 max ***v_c_*** = 15 m/min.

For the obtained sample surfaces after the turning process, selected GPS parameters were determined on an AltiSurf A520 machine (AltiSURF, Geneva, Switzerland) equipped with a CL2 confocal chromatic sensor with a working range up to 400 µm and a resolution of 8 nm in the optical axis of the device. The measurements were made on measuring areas of 2.0 × 5.0 mm along the radius of the tested cylinders. The scanning resolution was set as: along the x axis—0.47 µm, along the y axis—1.97 µm; this gave almost 110 million mapping points for each of the surfaces. Then the surface was divided into four parts depending on the cutting speed, resulting in test measuring areas of 2 × 1.25 mm. In Figure 2 the measurement areas are presented.

Digital processing of points for each test field and determination of the 3D surface topography parameters values were performed in AltiMap PREMIUM 6.2 software. Recorded point clouds were leveled (mean plane approximated by the method of least squares). Due to the high roughness of the sample after the SLM process, incorrectly collected points appeared during surface scanning (double reflections in “lenses” or bends at the edges) were removed by adapting the threshold values of the collected signal at the level of 0.05–99.95%, (deleted points were set as unmeasured values). The height parameters frequently used for parametric assessment of surface topography (in accordance with [20]), i.e., Sa (arithmetical mean height of the surface) and Sz (maximum height of the surface) were applied during the experiment. According to Królczyk et al. [21], the selection of these surface roughness parameters for an evaluation of a surface topography is important in terms of the parametric description of surfaces formed during the turning process.

The influence of the workpiece manufacturing process and cutting parameters on surface roughness parameters was examined by means of analysis of variance (ANOVA).

## 3. Results

### 3.1. Characteristics of the Tested Workpieces

#### 3.1.1. X-ray Diffraction Analysis (XRD)

Figure 3 presents the XRD spectrum obtained for the tested materials: cast samples and obtained in the SLM process.

The analysis of the phase composition by X-ray diffraction showed that both tested CoCr alloys (diversified in terms of fabrication method) have a similar phase composition. In both cases the dominant phase was γ-Co characterized by fcc (face centered cubic). Moreover, hcp (hexagonal close-packed) structure also was detected. This was due to the martensite transformation from the γ phase to the ε phase during the cooling process [2,6,22]. The visible difference in the intensity ratio of the highest γ and ε peaks for SLM sample may be due to the very rapid cooling during manufacturing process. Therefore, incomplete γ-ε phase transformation has occurred, which is clearly visible in the case of ε (201) peak, whose intensity is only slightly lower than γ (111). The additional phase consisting of M_23_C_6_ carbides ((M = Cr, Co, Mo) has also been identified, which stays in agreement with [7,22].

#### 3.1.2. Optical Microscope Analysis

Figure 4 shows the microstructure of the tested materials.

The tests showed that the sample made by a casting process (Figure 4a) has a dendritic structure, characteristic for cast CoCr alloys. The CoCr alloy has a microstructure consisting of an austenitic matrix, which is a solid solution of cobalt and chromium in the dendritic core structure. In crystallites, dendritic microsegregations, in which M_23_C_6_ type carbides are present, can be observed. The carbides formed tend to precipitate at grain boundaries and in interdendritic areas. For a sample produced by SLM (Figure 4b), arched molten pool boundaries are visible. It is also characterized by a certain porosity resulting from the presence of unmelted CoCr powder particles. On the micro scale, numerous small cell dendrites can also be seen. The observed microstructure corresponds to the descriptions contained in the literature for Co-Cr-Mo alloys [4,5,15].

#### 3.1.3. Vickers Hardness HV_0.1_ Measurements

The results of microhardness measurements for the tested samples are summarized in Table 4. The average values obtained for *n* = 12 measurement points are presented.

The obtained Vickers hardness values are similar and amount to 579 and 587 HV_0.1_ for samples after casting and the SLM process, respectively. Slightly lower values were obtained for CoCr castings; however, the distribution of obtained results is more uniform with a lower standard deviation of 21.12 HV_0.1_.

### 3.2. Materials Machinability Tests

#### 3.2.1. Measurements of GPS Parameters

The average values of *Sa* and *Sz* parameters recorded during the tests are presented in Figure 4. On the other hand, the values of the material ratio curve parameters, i.e., *Spk*, *Sk*, and *Svk* (Reduced Peak Height, Core Roughness Depth, and Reduced Valley Depth), used to identify the mechanism of surface irregularity formation during the cutting process are shown in Figure 5. Three measurement areas were taken from the samples produced, which gave 60 cases for each type of CoCr material (the results were registered in the test plan for 120 measurement fields). A design of experiment included the repetitions of measurements onto the neighbor areas of a surface instead the measurements on the other specimen. The location of three measurement areas was selected in such a way to provide the largest distance between them. This, to some degree, may deliver information on variability of the processing and does not increase the number of expensive SLM samples.

Based on the above results, a strong relationship between height (amplitude) parameters and technological cutting conditions can be seen. For both cast and obtained samples in the SLM process, along with the increase of machining parameters, the selected SGS parameters, i.e., *Sa* and *Sz*, are also increasing (a significant increase in the measured parameters for F4 and F5 in the case of casting and for F3 and F5 in the case of SLM is noticeable). The feed speed recommended by the manufacturer for the employed tools is in the range of 0.1–0.4 mm/rev, while the optimum is 0.3 mm/rev [23]. Analyzing the results obtained, it can be observed that for higher feed values (*f* ≥ 0.3 mm/rev), the obtained values of material ratio curve parameters increase, especially for higher recorded cutting speeds (V3 and V4). However, they are higher for samples after the SLM process. This may indicate an additional contribution of the factor associated with the workpiece material, which also participates in the mechanism of surface profile formation. In the case of tested samples, this may be the effect of the observed porosity of the material and/or the presence of unmelted CoCr powder particles, as well as the relatively high variability of microhardness of the analyzed material. It should be noted that variability of workpiece hardness can lead to changes in cutting force values, as reported in [24]. Consequently, the variable forces can induce vibrations in a machining system, influencing the surface topography formation. The relationship between the workpiece material and surface roughness can be more accurately determined by specifying the values of material ratio curve parameters such as *Spk*, *Sk*, and *Svk* (Reduced Peak Height, Core Roughness Depth, and Reduced Valley Depth)—(Figure 6).

Analyzing the dependence of the cutting parameters and the sample production method on surface topography, one can notice a less favorable course of material ratio curve parameters for the surfaces of CoCr samples made during the SLM process. The parameters of reduced roughness core depth *Sk* registered for both types of samples have a comparable course. However, in the case of the *Svk* parameter its significant increase is visible with increasing *f* value for samples obtained in the SLM process (Figure 6). A similar result can be observed for the *Spk* parameter—here also higher values were obtained for higher feeds in case of CoCr SLM samples. In practice, this can refer to different wear rates for parts made of CoCr by the SLM technique. Significantly lower tribological properties of products should be expected when the technologist uses high feeds and cutting speeds during machining. Clearances in cooperating parts manufactured in this way can be even higher by about 50% in relation to the clearance between parts for which cast semi-finished products were used. Sintered parts will wear out faster, especially in the final period of use. It is also anticipated that in the case of friction cooperation of such surfaces, their durability period is much shorter compared to materials obtained by classical techniques (metallurgy and foundry).

It is worth noting, that in case of average values of the *Spk* parameters in both tested CoCr samples, one can find that they are at a similar level (Figure 6). It can therefore be argued that the impact of the so-called material factor is even stronger at the higher values of the technological cutting parameters (usually feed) used during machining. Figure 7 gives exemplary isometric images of the machined surfaces with low feed rates (F2 = 0.15 mm/rev and cutting speed V1 = 60 m/min).

Surface isometric images of the samples after casting (Figure 7a) and SLM process (Figure 7b) confirm the observations regarding the GPS parameters (Figure 5 and Figure 6). Just as no significant quantitative differences were found for the GPS parameters determined, the similar nature of machined surfaces at low feed rates is noticeable (Figure 6). Thus, it can be argued that casts and samples fused with a laser beam have comparable machinability at low feed rates. However, considering the higher feed values (F3, F4, and F5), the characteristics of both GPS parameters and surface isometric images, in particular of SLM samples, are more diverse. The intensity of occurrence and the size of the observed defects on the surface increased with increasing feed value—Figure 8.

The presence of extensive and deep surface defects causes a significant increase in the values of GPS indicators, which can be caused on the one hand by the porosity of the material (removal of the material layer reveals the free spaces hidden under it), and on the other by the occurrence of significant friction forces and an increase in temperature between the surface of the tool and the workpiece subjected to plastic deformations [10,13,25]. It has also been observed that surface defects and some porosity of samples after SLM—that are exposed during cutting—are also partly rolled (Figure 7a). A very similar relation can be observed in surface topographies obtained during the roller burnishing process, as reported in [26,27,28]. During machining with high feed rates, a high value of the tangential force to the surface is generated, which results in exceeding the critical tangential stresses of the material, and in turn leads to chunking (tearing by the cutting edge) its large fragments (Figure 7b).

This aspect related to the machinability of parts produced from additively manufactured semi-finished products should be thoroughly understood and included in the mechanism of surface texture formation. In order to determine the place where a consistency loss of the machined material occurs, changes in GPS parameter values describing the shape of the material ratio curve were analyzed. By comparing the changes in the parameters describing the peaks, the core of the material and the depth of the valleys, one can accurately find the source of amplitude parameters. The multifactorial analysis of variance allowed the identification of observations significance and their potential source of origin.

#### 3.2.2. Analysis of Variance (ANOVA)

MANOVA (multivariate analysis of variance) was performed for the results of GPS parameter measurements obtained during experimental tests. It determined the significance of the impact of factors such as sample manufacturing technology (casting, laser beam melting, denoted as TECH), as well as kinematic and geometric factors used during cutting, i.e., feed per revolution (FEED) and cutting speed (SPEED). In order to maintain an accessible form of presentation of the obtained test results, the remainder of the work focuses only on the GPS parameters most often used in the description of the state of surface (as *Sa*, *Sz*) as well as ones enabling the determination of material aspects responsible for the mechanism of formation and development of observed quality defects, i.e., *Spk*, *Sk*, and *Svk*. Figure 9 shows the relationship of the main effects (TECH, FEED, SPEED) and the surface roughness parameters *Sa* and *Sz*.

The value of the lambda Wilks statistics obtained after conducting a multivariate ANOVA analysis clearly showed a significant relationship between the explained variables (GPS parameters) and the explanatory variables: Technology, Feed, Speed. The high relationship of interactions between all examined factors (1st and 2nd order) was also observed. Very interesting is the fact that the first-order interaction for the manufacturing technology is very strong with the state of GPS. The interaction of technology with feed and also the interaction of technology with cutting speed is much stronger than the observed main effect in the form of cutting speed alone. This relationship can be seen in a better way in the detailed results of one-dimensional ANOVA analysis—Table 5.

Among the factors considered, feed (*f*) had the most significant impact on GPS parameter values, followed by the interaction between the semi-finish product manufacturing technology and feed. The production technology had the greatest impact on the *Sz* and *Svk* parameter values, while the feed rate and interaction between them had a weaker effect. However, no significant impact of the semi-finish product manufacturing technology was found for the value of the core roughness depth (*Sk*) parameter. This means no contribution of material origin factor responsible for the formation of surface irregularities. The decisive values responsible for shaping the width of the surface core are only the kinematic and geometrical conditions of the machining process. Analyzing the contribution of individual parameters in the process of shaping the surface roughness, it can be seen that for the parameters *Sz* and *Svk*, the technology had the most important role, followed by feed and then the interaction between feed and speed. In the case of height (Amplitude) parameters and 3D functional parameters, such as *Spk* and *Sk*, they were most influenced by feed, then the interaction of technology and speed. In the above case, as for multivariate analysis of variance, the weakest impact of cutting speed was recorded, while the feed and its interactions with other technological machining parameters had the strongest impact. The feed had the most significant effect on machined surface peaks *Spk*. However, material manufacturing technology had the largest share in the generation of the reduced depth of valleys. First-order interactions, i.e., technology and feed, technology and cutting speed, and feed and cutting speed are illustrated in Figure 10.

Based on the analysis of variance, important technological recommendations can be made regarding the selection of machining parameters for parts made of CoCr alloys produced by casting or SLM technology. It turns out that during machining of these materials in a certain range of input parameters, no differences in GPS parameters were observed. Differences begin to appear during cutting with high volumetric machining efficiency. Appearing differences in the reduced valley depths (*Svk*), testify to the forceful impact of the cutting edge corner in machining of SLM samples, during which the burnishing of the workpiece by the tool flank face occurs, Moreover in progress of cutting time, the fragments of the surface are extracted (especially in the valleys). There are also significant differences at the irregularity peaks, which are more uniform for cast samples, and sharp (due to tearing) for samples produced with the SLM technique.

## 4. Discussion

The results of the phase composition tests by the XRD method, conducted for the CoCr alloys fabricated using casting and SLM indicated the presence of γ (fcc) and ε (hcp) cobalt phases in both tested samples [3]. According to [7,27] in case of proper alloying, the microstructure of the CoCr alloys is composed mainly of γ-phase and carbides of the M_23_C_6_ type. However, in numerous works concerning the cobalt-base alloys, their microstructure consists of a mixture of γ–Co and ε-Co phases [3,5,9,13]. It is dependent on solidification process and/or heat treatments applied to cobalt-base alloys due to the promotion of fcc→hcp transformation [5]. In the case of the SLM process, the phase transformation from fcc to hcp can occur thermally, by rapid cooling, which is induced by the local operation of laser beam on a small area, i.e., it melts the metal that solidifies quickly. The laser treatment induces a thermal phase transformation from γ (fcc) to ε (hcp) [9]. It is worth indicating that the effect of phase change is an increase of mechanical properties of these materials. The occurrence of γ-phase determines ductility, while the-ε phase has higher strength, and enhances the corrosion and wear resistance due to the limited number of slip systems [5,7,13].

Observations of the microstructure using optical microscopy showed that in both materials characteristic structures were obtained. The microstructure of traditional-cast specimen exhibited a typical dendritic solidification microstructure consisting of dendrites and an interdendritic region [4,5,8,13,15,29]. The SLM samples exhibit a peculiar macro-structure, similar to “fish scales”, relative to the weld pools of each laser pass and to layer stack [3]. Moreover, it reveals some porosity, which presents when the input energy is less than 150 J/mm^3^ [6]. Higher magnifications show that inside the “fish scales” a fine microstructure is present. Circular arch-shaped boundaries and numerous columnar grains can be observed, growing perpendicular to the melt pool boundary, because of heat transfer [3,6,15]. According to [3] an ultra-fine-grained material that is obtained by AM technique influences the very high strength of these specimens.

Microhardness tests using the Vickers method showed slightly higher (but also with a higher standard deviation) results obtained for samples obtained in the SLM process. In CoCr alloys in which cobalt occurs in two phases: ε is distributed as a network of thin lamellae inside the γ-phase. The higher hardness is attributed to the presence of the ε-lamellae, which restricts the movement of the dislocations in the γ-phase [7]. The higher hardness of samples, produced by SLM, is incident to the homogeneous microstructure with fine morphology and the higher volume fraction of the ε phase due to the incomplete γ-ε transformation, which is defined by the process character [19,30].

Referring to the results of surface topography measurements, it can be seen that for the cases analyzed, higher values of GPS parameters were obtained, during turning with higher feeds, similar to in [14,23,31]. It should be noted that according to [14,25,32], the higher feed rate can lead to higher tool wear and deteriorated surface finish. It is attributed to higher cutting force and higher temperature between tool and chip interface, which in turn contributes to the growth of friction, inducing the thermal softening of the cobalt binder phase and plastic deformation [10]. Adhesion wear and chipping are the dominant wear mechanisms operating at the rake and at flank faces when high feed rate is applied [32]. It can, therefore, be concluded that according to [9], the reduction of the feed causes the reduction in mechanical load of the cutting tool, and consequently an improved surface finish. A greater diversity of results obtained for laser-melted samples was also registered, which is in agreement with the results presented in [13].

Multivariate analysis of variance (MANOVA) showed that for most GPS parameters, feed was the most important factor, then the interaction between feed and the sample production technology, followed only by the method of obtaining samples. Only for one of the GPS parameters, i.e., *Sk*, the manufacturing technology had an insignificant statistical effect. In each case, the effect of cutting speed was negligible. The obtained results are consistent with other research regarding the relations between machining input parameters and surface topography [9,10,11,12,25,32], according to which, the feed has the greatest impact on the quality of the obtained surfaces.

## 5. Conclusions

In this work an evaluation of surface texture formed during turning of CoCr alloys fabricated by casting and SLM processes was made. Moreover, the conducted study involved the workpiece samples’ physical-mechanical characterization and multifactorial analysis of variance (MANOVA) related to machined surfaces, in order to evaluate the influence of technological cutting parameters and sample fabrication technique. On the basis of conducted studies, the following conclusions were formulated:The phase composition of the cobalt-chromium alloys obtained in casting and SLM processes was identical. The observed microstructures were characteristic for the manufacturing processes used. Moreover, the microhardness HV0.1 of materials obtained in the SLM process was higher than for cast samples; however, it was also characterized by a larger standard deviation.Calculated height (amplitude) parameters, i.e., *Sa* and *Sz* are characterized by a strong dependence on the technological cutting conditions. The registered GPS parameter values increase with increasing feed value. The observed influence of cutting speed on the change of GPS parameter values is much smaller. Analyzing the method of sample preparation, the quantitative changes occurring in CoCr alloy surfaces made by SLM method are much higher (higher values of GPS parameters and their variations were obtained, comparing to values reached for samples produced by a casting method).The recorded isometric images of the machined surfaces for both investigated samples confirm the observations regarding the GPS parameters—for low feed values a similar nature of machined surfaces is noticeable, which indicates that machinability of these materials is very similar. For higher values of feed, surface defects were observed, in particular for samples obtained in the SLM process, which caused a significant increase in the value of GPS indicators.The multivariate analysis of variance (MANOVA) showed that for the most of the analyzed factors, feed had the greatest impact on GPS parameters (*Sq*, *Sa*, *Spk*), followed by first order interactions between manufacturing technology and feed, and finally by the production technology. In case of *Sz* and *Svk* parameters, the production method had the highest impact, followed by the feed. Only in the case of *Sk* parameter, the impact of manufacturing technology was negligible. This allows to formulate the conclusion that features of surface topographies are mainly constituted during the machining process.When machining semi-finished products based on conventional metallurgical processes (forging, rolling, casting), high material removal rate can be achieved by using high feeds; however, in the case of semi-finished products manufactured by the SLM technique, a high machining performance can be reached by the application of high cutting speeds. In addition, during face turning of CoCr samples fabricated by the SLM technique, the feed values should not exceed the 0.15 mm/rev in order to assure the surface finish is comparable to one for the CoCr samples after the casting process. For selected cutting inserts, maximal, referenced by tool manufacturer, cutting speed is recommended. Feed ratio should be lowered to approx. 50% of maximal approved values (*Vc* = 45–60 m/min and *fz* = 0.1–0.15 m/rev).

## Figures and Tables

**Figure 1 materials-13-02448-f001:**
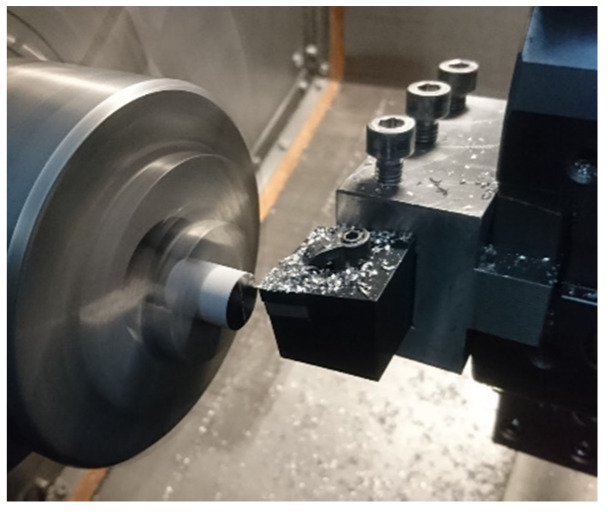
Face cutting process of a CoCr alloy sample on an AFM TAE 35 lathe.

**Figure 2 materials-13-02448-f002:**
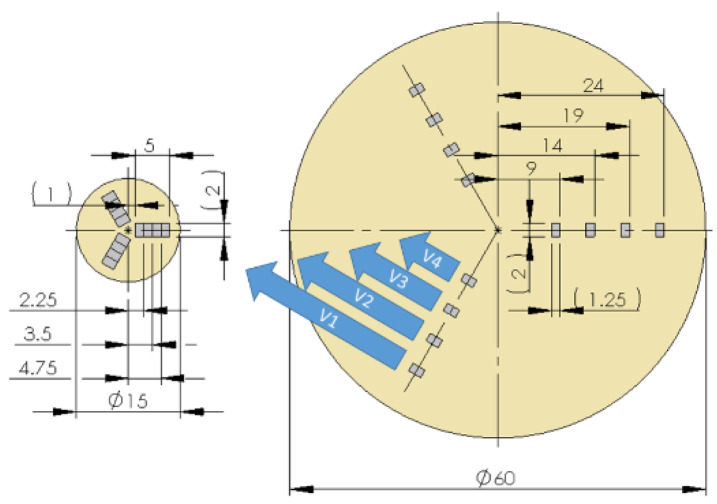
Measurements areas of the GPS analysis: small sample after the SLM process and larger sample after casting.

**Figure 3 materials-13-02448-f003:**
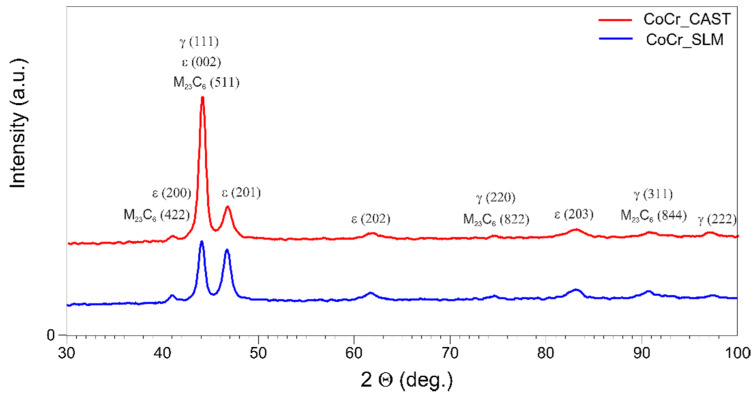
X-ray diffraction pattern of the tested samples.

**Figure 4 materials-13-02448-f004:**
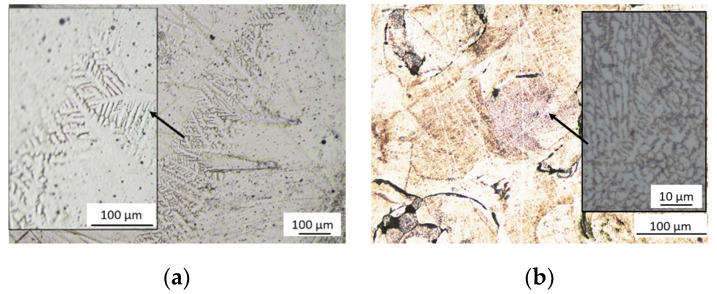
Optical microscope pictures obtained for tested sample: (**a**) cast sample; (**b**) SLM sample.

**Figure 5 materials-13-02448-f005:**
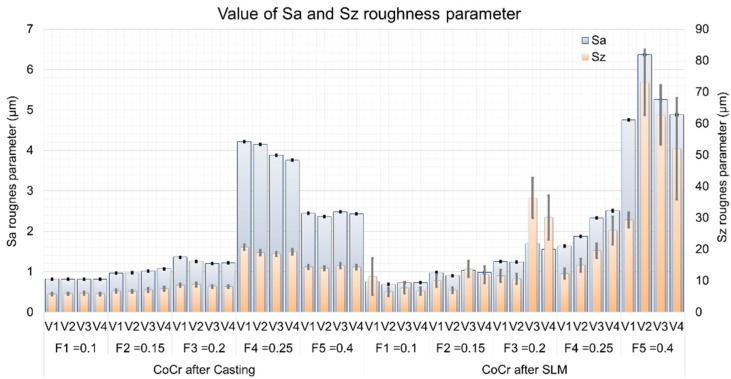
Comparison of *Sa* and *Sz* surface roughness parameters obtained for tested CoCr alloys and different *f* values.

**Figure 6 materials-13-02448-f006:**
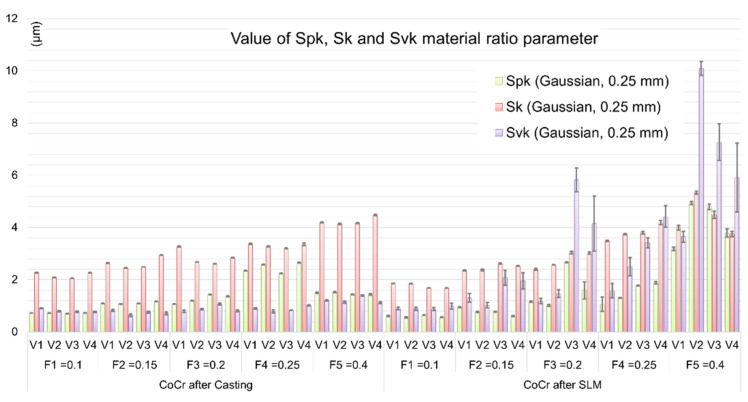
Comparison of material ratio curve parameters obtained for tested CoCr alloys and different *f* values.

**Figure 7 materials-13-02448-f007:**
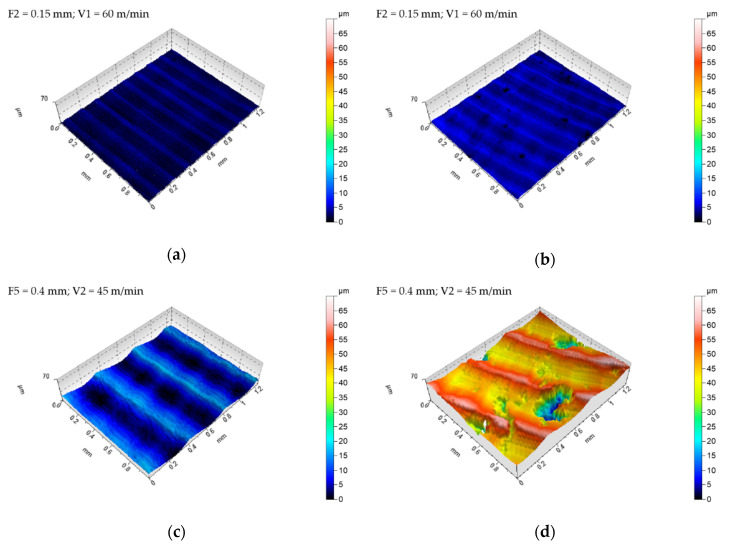
Isometric views of the surfaces machined with different technological parameters, i.e., F2 = 0.15 mm; V1 = 60 m/min and F5 = 0.4 mm; V2 = 45 m/min for: (**a**,**c**) cast sample; (**b**,**d**) SLM sample.

**Figure 8 materials-13-02448-f008:**
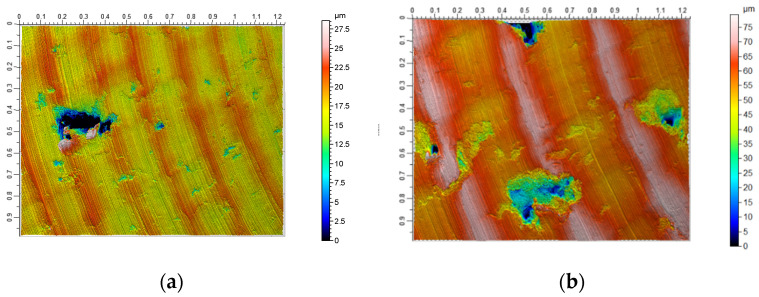
View of the surface of the samples obtained in the SLM process after face turning at higher values of feed: (**a**) F3 = 0.20 mm/rev; V3 = 0.5 *vc*max; (**b**) F5 = 0.40 mm/rev; V2 = 0.75 *vc*max.

**Figure 9 materials-13-02448-f009:**
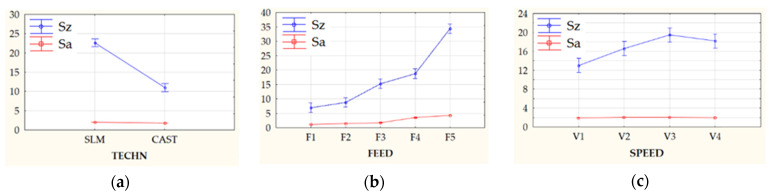
Changes in *Sa* and *Sz* parameters under the influence of main effects, where error bars indicate 95% confidence interval for the factor: (**a**) type of prefabricated element used—TECH, (**b**) feed speed—FEED; (**c**) cutting speed—SPEED.

**Figure 10 materials-13-02448-f010:**
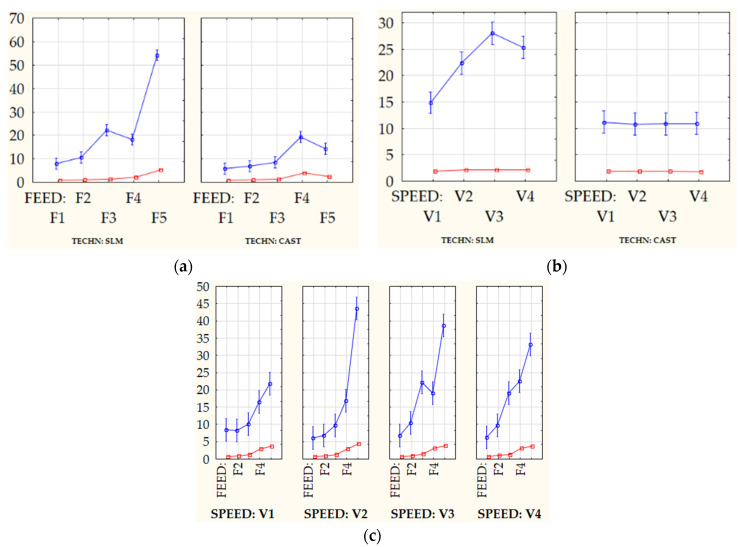
Significant interactions affecting the values of *Sa* and *Sz* parameters determined using multivariate ANOVA analysis where: (**a**) the impact of interaction between the semi-finished product technology and feed (TECH × FEED), (**b**) technology and cutting speed (TECH × SPEED) and (**c**) feed and cutting speed (FEED × SPEED).

**Table 1 materials-13-02448-t001:** Selective Laser Melting (SLM) process parameters.

Parameter	Value	Unit
Laser power/*P*	100	W
Scan speed/*V*	600	mm/s
Distance between laser paths/*h*	0.12	mm
Layer thickness/*d*	0.03	mm
Volume energy density/ε	99.21	J/mm^3^

**Table 2 materials-13-02448-t002:** Effective cutting tool (insert in tool holder) parameters.

Parameter	Value	Unit
Edge count	4	integer
Corner Radius	0.794	mm
Lead Angle	−6	deg
Clearance Angle	6	deg
Inclination Angle	−6	deg
Major Cutting Edge Angle	5	deg

**Table 3 materials-13-02448-t003:** Cutting parameters.

Pass Code	*f* (mm/rev)	*v_c_*—Var, Max in Area (m/min)	*a_p_*—Const (mm)
F1	0.10	60, 45, 30, 15	0.5
F2	0.15	60, 45, 30, 15	0.5
F3	0.20	60, 45, 30, 15	0.5
F4	0.25	60, 45, 30, 15	0.5
F5	0.40	60, 45, 30, 15	0.5

**Table 4 materials-13-02448-t004:** Results of Vickers Hardness HV_0.1_ measurements.

	Cast	SLM
Average	579	587
St. deviation	21.12	47.60

**Table 5 materials-13-02448-t005:** Detailed results of analysis of variance (ANOVA).

EFECT		SINGLE PARAMETER	TECH	FEED	SPEED	TECH * FEED	TECH * SPEED	FEED * SPEED	TECH * FEED * SPEED	ERROR	ALL
	df	1	1	4	3	4	3	12	12	80	119
Sz	SS	33920.40	4104.01	11392.16	696.01	6722.16	757.32	1821.80	1895.44	1319.93	28708.83
	MS	33920.40	4104.01	2848.04	232.00	1680.54	252.44	151.82	157.95	16.50	
	F	2055.90	248.74	172.62	14.06	101.86	15.30	9.20	9.57		
	C *	-	12.10	8.40	0.68	4.95	0.74	0.45	0.47		
Sq	SS	777.00	8.60	283.09	2.47	121.16	3.73	11.93	14.69	0.26	445.93
	MS	777.00	8.60	70.77	0.82	30.29	1.24	0.99	1.22	0.00	
	F	237466.76	2629.68	21629.53	251.68	9257.34	379.89	303.76	374.19		
	C *	-	1.11	9.11	0.11	3.90	0.16	0.13	0.16		
Sa	SS	483.21	1.24	181.29	0.38	71.11	0.86	2.15	3.89	0.01	260.94
	MS	483.21	1.24	45.32	0.13	17.78	0.29	0.18	0.32	0.00	
	F	5900370.90	15201.76	553427.87	1543.15	217082.21	3504.15	2191.79	3961.86		
	C *	-	0.26	9.38	0.03	3.68	0.06	0.04	0.07		
Spk	SS	295.31	3.22	71.86	2.25	47.31	2.04	5.47	3.72	0.25	136.11
	MS	295.31	3.22	17.96	0.75	11.83	0.68	0.46	0.31	0.00	
	F	94824.26	1035.05	5768.52	240.51	3798.05	218.32	146.33	99.41		
	C *	-	1.09	6.08	0.25	4.01	0.23	0.15	0.10		
Sk	SS	1108.82	0.00	80.04	0.23	2.84	1.80	2.53	3.38	0.03	90.86
	MS	1108.82	0.00	20.01	0.08	0.71	0.60	0.21	0.28	0.00	
	F	2545543.77	0.26	45935.63	179.71	1627.52	1379.22	484.91	646.20		
	C *	-	0.00	1.80	0.01	0.06	0.05	0.02	0.03		
Svk	SS	475.22	140.92	141.50	20.24	103.46	20.05	45.24	40.53	2.30	514.24
	MS	475.22	140.92	35.38	6.75	25.87	6.68	3.77	3.38	0.03	
	F	16536.59	4903.79	1230.99	234.74	900.07	232.56	131.19	117.52		
	C *	-	29.65	7.44	1.42	5.44	1.41	0.79	0.71		

* Contribution [%].

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
