# Peer review of "Evaluation of Surface Topography after Face Turning of CoCr Alloys Fabricated by Casting and Selective Laser Melting"

_materials, 2020, doi:10.3390/ma13112448_

Round 1

Reviewer 1 Report

This work focuses on the evaluation of surface texture formation during face turning of CoCr alloy fabricated by casting and selective laser melting. The present paper is interesting, however, to be accepted for publication the following comments need to be addressed

  • The introduction section needs to be improved by citing new and related articles of the current journal.
  • The specialized word should be unified. Such as "Figure" and "Fig" (some but not all) in L 116, L 121,…….
  • In Figure 3, the scale bar isn’t clear. This figure should be improved.
  • In ANOVA part, I recommend putting a table to summarize the controlled parameters and the levels of each parameter. In addition, in the results of the ANOVA analysis, the table should contain the Rank and contribution of each parameter.

Reviewer 2 Report

The authors provide a study on the surface properties of SLM manufactured CoCr alloys after face turning, which could be a promising alternative to cast metal given the high hardness and elasticity and bad machinability.

The samples were investigated after face turning, clearly the roughness of SLM manufactured samples increases faster with rotational speed compared to cast samples.

However, whilst face cutting and the machinability of SLM samples would probably not be the property of highest significance with respect to the later use (since they could be pre-produced by SLM near to any design needed for the application), it provides possible insight on mechanical properties of SLM-manufactured samples and durability.

The authors might point out the findings of their investigations in more general terms as resistance to sheer and bending forces, that could be directly applicable in design of applications.

Reviewer 3 Report

The manuscript is very interesting. I think that it needs no corrections. 

Reviewer 4 Report

This paper focuses on evaluation of surface texture formation during face turning of Co-Cr alloy. The specimens analysed are realized by means of the application of conventional casting or manufactured by additive manufacturing - SLM. Different analytical and experimental methods were employed to describe the specimen composition and morphology, as well an ANOVA analysis have been developed to verify the influence of manufacturing technology, cutting speed and feed ratio on selected surface parameters. Finally, the authors verified that in a range of higher feed values, the surface quality of Co-Cr samples fabricated by SLM was lower than that reached for Co-Cr after casting process.

Although the paper is interesting for future readers, verifying the machinability of hard-to-cut Co-Cr alloys manufactured by Selective Laser Melting (SLM) technology and providing the results of this analysis, I believe that the authors can greatly still improve their paper. In fact, I believe that the introduction has been developed correctly and in general the structure of the work is the correct one. But there are points that should be improved. The conclusions of the work don’t convince me, the research project itself could be explained better and more clearly, explaining which characteristics the Co-Cr components must have for the application for which the authors believe they can replace the conventional casting process. There are shortcomings in relation to the experimental methods and parameters that need to be filled, both as regards the SLM manufacturing process and the turning process, as well as nothing is said about the casting process. In turning machining, the definition of the process window is unclear, as well as how the analysis factors and their levels have been chosen. At the level of experimentation, there are no replicas of the processes; therefore it is not possible to have precise information on the actual replicability after (and also before) the turning process, but only on the differences in the roughness parameters between neighboring areas. Even the initial roughness value of the specimens is not known, for either of the two production technologies (SLM and foundry), therefore it is difficult to evaluate the increase in surface finish obtained.

The statistical treatment is very long, full of output data (perhaps too many), and it would have been expected that it would actually lead to accurate process optimization conclusions; instead the authors have drawn only a few useful conclusions.

Below are detailed all the points to which the authors should carefully address:

  1. At lines 70-71, the authors refer to the disadvantages of the casting process, without specify which are, but only inserting a citation. The authors should clarify better this point, explaining, also in the text, how the SLM process manages to avoid these particular problems.
  2. At line 103, about table 1, it is not clear how and why the SLM process parameters have been chosen by the authors, or if they are process values imposed by the SLM machinery.
  3. At lines 111-112, the authors should explain according to which logic the indentations were made, the reason for choosing 12 tests and at which distance between them. These considerations should be present also in the text. Maybe a drawing would have facilitated the understanding of the reader.
  4. At lines 115-116, the authors refer to the basic parameters of the inserts, but they don’t explain how they have been chosen and if there is a reason behind this choice. The authors should improve also the text, explaining better this point.
  5. At line 118, about table 2, I don’t understand the reason for including also the “edge count” in the table. Is it an important parameter of the experimentation?
  6. At line 122, the authors explain that “Machining was carried out without cutting fluids”, but they don’t explain the reason behind this choice and if it is normal for the machining of CoCr alloys.
  7. At line 127, about table 3, it is not explained how the process window have been chosen, or according to what logic these values were and not others. The authors should also clarify the reason of choosing constant rotational speed and cutting depth, as well as variable cutting speeds. All these fundamental information should be also inserted in the text.
  8. At lines 135-137, the authors report that “measurements were made on measuring areas of 2.0 × 5.0 mm along the radius of the tested 136 cylinders”. Is this area sufficient to cover the entire machining area and to represent possible different morphological behaviors at different depths?
  9. At line 156, about figure 2, the unit of measurement and the scale of the signal “intensity” count are not present on the graph, while they should be there.
  10. About lines 162-164, the authors should better clarify the reason of the different behaviors of SLM and casting samples. I don’t understand why for the SLM sample the first peak is much lower than the casting sample, while the second is much higher. From reading the text it is not clearly understandable what this result is attributed to. So it should be explained much better.
  11. At line 177, the measurement scales of figures 3a and 3b are too little and difficult to read. They should be bigger to be legible.
  12. Observing the results of table 2, line 182, I continue to think that should be important to explain which the indentation points are and where are they positioned. As well as, why the standard deviation of the Vickers hardness after SLM sample is more than double that after the foundry process. The author should also provide a clear explanation in the text.
  13. Starting from line 189 onwards, the authors report the results of the surface roughness analysis, for the various specimens analyzed. I wonder why they did not carry out this analysis also on the specimens deriving from the casting and the SLM processes, therefore before turning. Without those parameters, neither measured nor reported from the literature, it is difficult to understand the surface finishing improvements by turning. It would be advisable for the authors to enter them, both average values and standard deviations, so that at least the reader understands if the overall pass depth of the turning process is sufficient to eliminate any initial defects present or if these can be propagated in subsequent finishing processing.
  14. At lines 193-194, the authors report that “Three measurement areas were taken from the samples produced”. So the authors did not replicate the process, but only chose the neighboring areas between them. This, in my opinion, does not give sufficient indications on the repeatability of the processing, between one specimen and the next, but only provides indications regarding the variability of the quantities on the single specimen. The overall standard deviation of the quantities, in the reporting of the tests, will therefore in all probability be greater than that estimated by the authors.
  15. In addition to this, I would like to know how the three measurement areas on the same specimen were chosen, since it is not specified in the text and should be clarified.
  16. At line 196, about figure 4, I don’t see any indication of Sa standard deviation for each specimen… why isn't it present in the chart? Also in the text it should be explained what values it assumes if it was actually measured. If it wasn't, the authors should clarify why.
  17. At line 196, about figure 4, I can see a particular trend both for the casting process (F4 and F5 generate Sz, Sa much higher than the others) and for the SLM process (F3 generates significantly higher values than F1 and F2, then the values go down on F4, to go up a lot on F5). It seems to me that all this is not underlined in the text, but only in the final discussion (chapter 4) and I don't understand why, since it is immediately evident. The text should therefore be implemented according to a more accurate analysis of the charts.
  18. At lines 210-212, the authors have written that “In the case of tested samples, this may be the effect of the observed porosity of the material and/or the presence of unmelted CoCr powder particles, as well as the relatively high variability of microhardness of the analyzed material”. Why the high variability of microhardness of the analyzed material should influence the roughness parameters? The authors should clarify this point better, also in the text.
  19. At line 238, about figure 6a and 6b, why didn't the authors use the same scale for both charts? The differences between the two maps would have been more clearly understood by the reader.
  20. At the end of the line 240, the authors refer to a figure 3b, but it seems to me to be wrong. Maybe was figure 6b the correct one?
  21. At line 248, about figure 7a and 7b, it would have been very useful to add 3D maps of foundry specimens, worked on the lathe with the same parameters, so as to compare the results of the different machining analysed.
  22. At line 279, about figure 8, I don’t understand why the lines are not well aligned with the scales at the bottom, for all the graphs. The authors should correct this non-alignment.
  23. At page 10, several times, lines numbers are superimposed on the text. I don't know it's a formatting problem or what, but the effect is quite unpleasant, since it also makes reading not so simple.
  24. At page 11-12, the entire discussion is in my opinion too long. There are many repetitions compared to what is already reported in chapter 3 of the text. Furthermore, too much space is given to analyzing not so significant concepts. The authors should slightly simplify the discussion, focusing on the analysis of the most significant results of the various analyzes and uniting them better together, in an overall analysis that is less fragmented and more connected in the various aspects.
  25. About both the abstract of page 1 and the conclusions of page 12, the authors should explain the why of the importance of the following concept: “However, in a range of higher feed values, the surface quality of CoCr samples fabricated by SLM was lower than that reached for CoCr after casting process”. That is, why are these roughness values a problem? What applications do not make it possible? Are there any limit values to be respected? I believe that on this point, even the research project could be better explained to the reader, who hardly understands why these values are not acceptable, if the precise case study at application level is not well specified.
  26. About line 497-498, the authors should better clarify the self-citation number [18], inserted at line 148. Reading the text at line 146-148, the link between the sentence and the reference is not clear. Then the text should be integrated, specifying better what the citation that has been inserted in the work refers. Otherwise it seems that its insertion took place very forcedly.

Round 2

Reviewer 4 Report

In the second version of the paper, the authors greatly improved their paper, managing to correct most of the shortcomings highlighted in the first revision. The research project is now better explained, as well as experimental methods and parameters. The choice of the turning process window is now clearer. The conclusions have been reduced and are certainly simpler, clearer and more effective. Despite this, the work still has some weaknesses, some of which the authors could have easily implemented. The absence of processing replicas does not allow to know the actual repeatability of the turning processing, but only the variation of the roughness parameters on neighboring areas of the same specimen. The initial roughness would have been appropriate to measure and has not been done. Reporting some considerations about the optimization of the process parameters would have been appropriate too. But the fundamental point that remains unclear to me concerns the roughness areas analyzed for both technologies. Figure 2 and the comments added by the authors only complicated the understanding of characterization methods, which should be described really much better.

In summary, the authors provided answers to all the points that had been listed, but on some of these there are some aspects that should be clarified in a much greater way. So in detail:

  1. The statistical treatment is very long, full of output data (perhaps too many), and it would have been expected that it would actually lead to accurate process optimization conclusions; instead the authors have drawn only a few useful conclusions.

Response: So far evaluation of surface quality is carried out visually by an experienced dental technician. In the paper, we attempted to relate GPS parameters with manufacturing technology. One of the objectives of the paper was to select GPS parameters sensitive to processing parameters.

Reviewer: What the authors reported is correct, but they could still define an optimal process window for the SLM machining, after having characterized the process. It would only increase the applicative contribution of the paper, according to the case study analyzed.

  1. At line 122, the authors explain that “Machining was carried out without cutting fluids”, but they don’t explain the reason behind this choice and if it is normal for the machining of CoCr alloys.

Response: The reason for “dry machining” was independent of technological recommendations. Cutting force gauge was installed in tool shank. Cutting forces were measured during the process and therefore cooling fluid was not applied, although technologically recommended.

Reviewer: I do not find reference in the text to the shear forces measured during the process ... why were they measured? To get which information? This concept should also be explained in the text.

Furthermore, if the use of lubricant is technologically recommended for this process, it should be specified in the text, to also inform the reader about this practice.

  1. At lines 135-137, the authors report that “measurements were made on measuring areas of 2.0 × 5.0 mm along the radius of the tested 136 cylinders”. Is this area sufficient to cover the entire machining area and to represent possible different morphological behaviors at different depths?

Response: Figure 2 which presents the measured areas was added to the manuscript.

Reviewer: Figure 2 does not clarify my doubts about the understanding of the measurement, indeed quite the opposite; perhaps it is the most delicate point of this second review. If the turning processes start from different diameters of the specimens produced via SLM and casting process, respectively 15 and 60 mm, how is it possible to have the same areas measured in this way?

The explanation in the text in the new lines 181-188 is far from clear, the figure on the basis of this explanation is confused, how can the areas measured for the two processes be the same? Given the different diameters, how were the areas with higher speeds V1, V2 and V3 evaluated for the SLM? The authors must provide a much clearer explanation than the current one, which tends to confuse the reader.

  1. Observing the results of table 2, line 182, I continue to think that should be important to explain which the indentation points are and where are they positioned. As well as, why the standard deviation of the Vickers hardness after SLM sample is more than double that after the foundry process. The author should also provide a clear explanation in the text.

Response: A partial answer to the question is included in point 3. In  samples,  produced  by SLM,  the  higher hardness  is  attributed  to  the  homogeneous microstructure  with  fine  morphology  and  the  higher volume fraction of ε phase due to the incomplete γ-ε transformation, which is defined by the process peculiarities [7].

Reviewer: Where are these considerations in the text? Why weren't they included?

  1. Starting from line  189  onwards,  the  authors  report  the  results  of  the  surface roughness analysis, for the various specimens analyzed. I wonder why they did not carry out this analysis also on the specimens deriving from the casting and the SLM processes, therefore before turning. Without those parameters, neither measured nor reported from the literature, it is difficult to understand the surface finishing improvements by turning. It would be advisable for the authors to enter them, both average values and standard deviations, so that at least the reader understands if the overall pass depth of the turning process is sufficient to eliminate any initial defects present or if these can be propagated in subsequent finishing processing.

Response: Both types of specimens were prepared for finishing processing analyzed in the paper. Facing prior to investigated turning was carried out to provide planar surface. We also assumed that surface roughness of all specimens was at the similar level. Measurement of roughness parameters would require removing the specimen from the lathe and then mounting the workpiece to the spindle and repeated planar surface preparation.

Reviewer: the authors assumed that the initial roughness values were the same, but it would have been appropriate to measure them. It was just a matter of completing a measurement on an additional specimen for each technological process. It was definitely a sustainable effort compared to the rest of the paper. In absence of these measurements, at least one reference value reported from scientific literature about rough specimens for each technology would be appropriate to provide.

  1. At lines 193-194, the authors report that “Three measurement areas were taken from the samples produced”. So the authors did not replicate the process, but only chose the neighboring areas between them. This, in my opinion, does not give sufficient indications on the repeatability of the processing, between one specimen and the next, but only provides indications regarding the variability of the quantities on the single specimen. The overall standard deviation of the quantities, in the reporting of the tests, will therefore in all probability be greater than that estimated by the authors.

Response: Indeed our design of experiment included the repetitions of measurements onto the neighbour areas of a surface instead the measurements on the other specimen. We agree that the best solution is fabrication of additional specimens to replicate the process. Nevertheless the location of three measurement areas was selected in such a way to provide the largest distance between them. This, to some degree, may deliver information on variability of the processing and does not increase the number of expensive SLM samples.

Reviewer: What the authors report is quite correct, certainly the costs are minimized thanks to this solution. But in this way the only information obtained is about the variability of the roughness parameters linked to the single process, that is to the single sample. The repeatability of the process is not known in this way, as is the variability between consecutive samples, certainly greater than that found on the single sample.

In the text the authors should report some of the comments of this answer, to underline how the values of variability provided are related to those of the single specimen, not to the processing of consecutive specimens.

  1. At line 238, about figure 6a and 6b, why didn't the authors use the same scale for both charts? The differences between the two maps would have been more clearly understood by the reader.

Response: It has been corrected in the revised manuscript.

Reviewer: the angle of the old figures 6a and 6b was preferable to the current one of figures 7 (a-b-c-d), since it allowed to better appreciate the surface texture of the specimens and any defects. I also recommend always using the same angle for all figures, for a more profitable comparison between them.
